# Association of Maternal Plasma Manganese with the Risk of Spontaneous Preterm Birth: A Nested Case–Control Study Based on the Beijing Birth Cohort Study (BBCS) in China

**DOI:** 10.3390/nu15061413

**Published:** 2023-03-15

**Authors:** Weiling Han, Wei Zheng, Aili Wang, Junxi Chen, Jia Wang, Junhua Huang, Hang An, Yuru Ma, Kexin Zhang, Ruihua Yang, Lailai Yan, Zhiwen Li, Guanghui Li

**Affiliations:** 1Division of Endocrinology and Metabolism, Department of Obstetrics, Beijing Obstetrics and Gynecology Hospital, Capital Medical University, Beijing 100026, China; 2Beijing Maternal and Child Health Care Hospital, Beijing 100026, China; 3Department of Obstetrics, Beijing Luhe Hospital, Capital Medical University, Beijing 101100, China; 4Institute of Reproductive and Child Health, Peking University, Beijing 100191, China; 5Key Laboratory of Reproductive Health, National Health Commission of the People’s Republic of China, Beijing 100191, China; 6Department of Epidemiology and Biostatistics, School of Public Health, Peking University, Beijing 100191, China; 7Department of Laboratorial Science and Technology, School of Public Health, Peking University, Beijing 100191, China

**Keywords:** manganese, spontaneous preterm birth, repeated measurement, dynamic monitoring

## Abstract

We performed this study to clarify the dynamic changes in maternal manganese (Mn) concentration during pregnancy and its association with spontaneous preterm birth (SPB). A nested case–control study was conducted based on the Beijing Birth Cohort Study (BBCS) from 2018 to 2020. Singleton pregnancy women aged 18–44 (*n* = 488) were involved in the study, including 244 cases of SPB and 244 controls. All of the participants provided blood samples twice (in their first and third trimesters). Inductively coupled plasma mass spectrometry (ICP-MS) was used for the laboratory analysis, and unconditional logistic regression was used for the statistical analysis. We found that the maternal Mn levels were significantly higher in the third trimester than those in the first trimester (median: 1.23 vs. 0.81 ng/mL). The SPB risk was increased to 1.65 (95% CI: 1.04–2.62, *p* = 0.035) in the highest Mn level (third tertile) in the third trimester, especially in normal-weight women (OR: 2.07, 95% CI: 1.18–3.61, *p* = 0.011) or non-premature rupture of membrane (PROM) women (OR: 3.93, 95% CI: 2.00–7.74, *p* < 0.001). Moreover, a dose-dependent relationship exists between the SPB risk and maternal Mn concentration in non-PROM women (P trend < 0.001). In conclusion, dynamic monitoring of maternal Mn level during pregnancy would be helpful for SPB prevention, especially in normal-weight and non-PROM women.

## 1. Introduction

Preterm birth, defined as a live birth between 28 and 37 weeks of gestation, is the leading cause of perinatal death and the death of children under five years old worldwide, with an incidence of 6.4% [1,2]. Preterm birth can be classified into spontaneous and iatrogenic preterm birth; the former accounts for from 70% to 80% of total preterm births [1,2]. In contrast to iatrogenic preterm birth, which has a relatively specific cause, the causes and mechanisms that result in spontaneous preterm birth (SPB) remain unclear.

Manganese (Mn), as one of the essential trace elements, may play a vital role in fetal growth and development [3,4,5,6]. A national nutrition and health survey in the United States showed that the blood Mn concentration of pregnant women was higher than that of non-pregnant women, suggesting that Mn plays an essential role in pregnancy [7]. The adequate intake (AI) of Mn during pregnancy recommended by the Institute of Medicine (IOM) is 2 mg/day [8]. In comparison, the World Health Organization and the Food and Agriculture Organization (WHO/FAQ) designated 5 mg/day as the recommended dietary intake (RDI) [9]. We are exposed to Mn through diet (such as drinking water, leafy vegetables, nuts, grains, and tea) and environment (the respiratory inhalation of vehicle exhaust, skin absorption, etc.) [10]. As a result, Mn deficiency has been extremely rare in recent years, and there are no reports of Mn deficiency in non-experimental conditions. Conversely, environmental exposure is making an excess of Mn increasingly common. The harm that an excessive amount of Mn can cause to health is becoming gradually apparent. Labor (term and preterm) onset is characterized by increased myometrial contractility, cervical dilatation, and rupture of the chorioamnionitis membranes [11]. Inflammation and oxidative stress play essential roles in the process [11]. As molecular mechanisms of Mn toxicity include oxidative stress [12], maternally inappropriate Mn concentrations may be involved in the labor onset of SPB.

Several studies have discussed the relationship between Mn concentration and preterm birth. However, no consistent conclusions were made. Studies conducted in different countries suggested a positive correlation between Mn levels and preterm birth [13,14,15]. However, a study based on the Japan environment and children’s study (JECS) did not find relationships between high-level Mn exposure and an increased risk of premature birth [16]. Similarly, a study in Bangladesh also failed to conclude that Mn is related to the risk of premature birth [17]. These conflicting results may be due to ethnicity, region, exposure level, and sample time. Moreover, previous studies did not strictly distinguish between SPB and iatrogenic preterm birth, leading to a confusing conclusion. In addition, the sampling times used in existing studies were a single point in time, most of which occurred at delivery, and dynamic monitoring data during pregnancy were lacking.

Therefore, this study explores the relationship between the maternal plasma Mn level and SPB through repeated measurement data to discover SPB’s underlying drivers in micronutrient views. The measurement is carried out in the first and third trimesters of pregnancy. We aim for this study to be helpful for clinical prevention and intervention measures for preterm birth to reduce the occurrence of preterm birth and improve the quality of the birth population.

## 2. Materials and Methods

### 2.1. Study Design and Participants

A nested case–control study was conducted based on a prospective maternal and child health cohort, the Beijing Birth Cohort Study (ChiCTR2200058395), in Beijing, China. In total, 32,496 pregnant women were recruited in the cohort from 2018 to 2020 at Beijing Obstetrics and Gynecology Hospital, Capital Medical University. We recruited women with the following inclusive criteria: aged 18–44 years old, with gestational weeks ≤ 14 weeks during recruitment, with no mental illness, and who could provide signed informed consent. Participants were examined and followed up until delivery. We collected maternal and fetal information prospectively when participants came to the hospital for an antenatal examination, and the information was recorded in the electronic clinical system. The ethics committee of Beijing Obstetrics and Gynecology Hospital, Capital Medical University, approved the study (2018-ky-009-01).

Figure 1 depicts the screening process. Out of the 32,496 pregnant women in the cohort, excluding those who were lost to follow-up (*n* = 946), stillbirth (*n* = 46), fetal defects (*n* = 968), multiple pregnancies (*n* = 1030), and cervical insufficiency (*n* = 203), a total of 29,303 live singleton pregnant women with complete medical records remained, including premature births (*n* = 1599) and term births (*n* = 27,704). Among the preterm birth group, a final amount of 244 pregnant women were included in the SPB group after excluding iatrogenic preterm birth (*n* = 982) and a lack of plasma samples in the first or third trimester (*n* = 373). In the full-term birth group, women were excluded if they met the following criteria: gestational weeks were <39 weeks or >41 weeks (*n* = 8008), if they experienced pregnancy complications (*n* = 11,949), or if there was a lack of plasma samples in the first or third trimester (*n* = 2567). Then, the control group was randomly selected according to the blood sampling time (1:1).

### 2.2. Definitions

In our study, preterm birth was defined as a live birth during the gestational ages of 28–36 + 6 weeks. SPB was defined as the spontaneous onset of labor at a gestational age of 28–36 + 6 weeks, including premature birth with a partial premature rupture of membranes and intact membranes. Iatrogenic premature birth refers to cases that require a medical intervention to give birth at 28–36 + 6 weeks. One reason for premature birth may be the mother’s or fetus’ poor health status that did not allow for pregnancy to continue (continuous deterioration of maternal pregnancy complications, infection, fetal distress, etc.). The gestational week was checked according to the last menstrual period (LMP), the course of pregnancy, and ultrasound measurement data in the first trimester. The first trimester referred to gestational weeks < 14 weeks, and the third trimester referred to gestational weeks ≥ 28 weeks. Body mass index (BMI) was classified into three grades based on expert consensus on medical nutrition treatment for overweight/obesity in China (2016) [18]. The criteria were as follows: slim: BMI < 18.5 kg/m^2^; normal: 18.5 ≤ BMI < 23.9 kg/m^2^; and overweight: BMI ≥ 24 kg/m^2^.

### 2.3. Laboratory Analysis

Plasma samples were chosen for Mn level determination to reflect nutrition intake to a certain degree. We collected plasma samples from the same pregnant women in their first and third trimesters. Maternal venous blood samples were collected by healthcare workers after 8–10 h of fasting. Then they were placed into vacuum blood collection vessels lined with sodium ethylene diamine tetraacetic acid (EDTA). All blood samples were centrifuged at 1680× *g* for 10 min within 1 h after collection, and plasma samples were extracted and stored at −80 °C until analysis. Inductively coupled plasma mass spectrometry (ICP-MS, 7700×, Agilent, Palo Alto, California, CA, USA) was used to detect plasma Mn concentration. A plasma sample of 0.1 mL was added to a 2 mL centrifuge tube. Then 0.1 mL of internal standard indium (In) (0.2 ng/mL, National Nonferrous Metals and Electronic Materials Analysis and Test Center, Beijing, China) and 1.8 mL of 1% nitric acid (UPS grade (68%), Suzhou Crystal Clear Chemical Co., Ltd., Suzhou, China) were added into the 2 mL tube. All of the samples were shaken sufficiently before determination. The median measured concentration of standard plasma samples (ClinChek^®^Plasma Control, Level II: 8884) was 15.3 ng/mL, which was within the range of the reference value: 15.2 (12.2–18.3) ng/mL. The detection limit of Mn was 0.008 ng/mL. We conducted the quantitative analysis in the Central Laboratory of biological elements of the Health Science Center of Peking University. The program has passed the China metrology certification system. Specific detection methods have been described in previously published studies [14].

### 2.4. Sample Size Estimation

According to a previous search of the literature, the exposure rate of high-level Mn in the term birth group was about 50%. The SPB risk was significantly increased to 2.46 (95% CI: 1.08–5.62) at the highest level of Mn in the first trimester [14]. Assuming that the risk-odds ratio is 2, the significance level of the bilateral test is α = 0.05, β = 0.1. The SPB group sample size was at least 179, calculated using the PASS2021 software (PASS 2021 Power Analysis and Sample Size Software (2021). NCSS, LLC. Kaysville, Utah, UT, USA). As a 1:1 case–control study design was used here, 244 women in the SPB group and 244 women in the term birth group were eventually selected for final analysis.

### 2.5. Statistical Analysis

Student’s *t*-tests or Pearson’s chi-square test were used for the comparison of the maternal baseline information. As a result of the skewed distribution of Mn concentration, the median with an interquartile range (IQR) was used to describe Mn exposure level. The Mann–Whitney U test was used to compare the two groups’ differences. Wilcoxon’s signed-rank test was used to compare the difference in Mn concentrations between the first and third trimesters. In the dose–response analysis, we divided all women into three equal groups based on the Mn concentration: low, medium, and high. Unconditional binary or multiple logistic regression was used to analyze the relationship between Mn level and SPB risk in different populations with or without adjusting for confounders. The odds ratio (OR) and a 95% confidence interval (CI) were used to quantify the SPB risk. Adjusted confounders included age, BMI, education, economy, nationality, gravida, parity, sampling time, and fetal gender. All the confounders involved have been reported to have effects on preterm birth [19,20,21]. As inflammation may affect SPB, we screened out factors in the 1st and 3rd trimester blood routine test that may affect SPB as inflammatory confounders. Univariate logistic regression was used in this process. Stratified analyses were carried out according to pre-pregnancy BMI and parity. We also conducted a sensitivity analysis to verify the stability of the statistical results. Women with vaginal group B streptococcus (GBS) infection (*n* = 9 in the SPB group and *n* = 16 in the term labor group) were excluded from the sensitivity analysis. Spss26.0 (Released 2019. IBM SPSS Statistics for Windows, Version 26.0, IBM Corp, Armonk, New York, NY, USA) was used for data analysis, and a two-tailed *p*-value < 0.05 was considered statistically significant. Regarding multiple testing by analyzing various subgroups, Bonferroni correction was used here to adjust the significant *p*-value [22].

## 3. Results

### 3.1. Baseline Information

In this study, a total of 488 women were included in the final analysis, including 244 cases (SPB) and 244 controls (term birth). The primary characteristics of the mothers and fetuses are summarized in Table 1. All of the women were non-smokers. The average age of all the participants was 31.51 ± 3.91 years old. The SPB group had a slightly higher average BMI and a higher proportion of overweight women (25% vs. 13.1%, *p* = 0.003). The SPB group had a lower proportion of nulliparous than the controls (64.3% vs. 76.2%, *p* = 0.004). All of the women in the SPB group were late preterm (34–36 + 6 weeks), and the cesarean section rate was higher than that in the term labor group (39.8% vs. 27.2%, *p* = 0.003). There was a similarity between the two groups in terms of education, economy, nationality, sampling time (8 weeks in the first trimester and 34 weeks in the third trimester), and fetal gender. All women were not antibiotic-treated during pregnancy.

### 3.2. Mn Concentration in the First and Third Trimesters

The median plasma Mn concentration for all women was 0.81 ng/mL (IQR: 0.63 ng/mL) in the first trimester, and it was 1.23 ng/mL (IQR: 0.67 ng/mL) in the third trimester. Despite the fact that there was a slightly higher Mn median concentration in the SPB group than that of the controls, it was not statistically significant (0.82 vs. 0.79 ng/mL, *p* = 0.441 in the first trimester; 1.27 vs. 1.20 ng/mL, *p* = 0.119 in the third trimester). When comparing Mn levels between the first and third trimesters, a cumulative effect during pregnancy was found, as the Mn concentration in the third trimester was significantly higher than that in the first trimester (*p* < 0.001) (see Table 2).

### 3.3. Relationship between Mn Level in the Third Trimester and SPB Risk

Table 3 displays the association between Mn levels and SPB risk. We divided all the women into three equal groups based on Mn concentration for the dose–response analysis. Overall, we found no significant stable relationship between the plasma Mn level in the first trimester and SPB risk. SPB risk was increased to 1.54 (95% CI: 1.00–2.39, *p* = 0.052) in the highest Mn concentration in the third trimester. After adjustment for clinical confounders, including age, BMI, education, economy, nationality, gravida, parity, sampling time, and fetal gender, the association became significant (OR: 1.60, 95% CI: 1.02–2.53, *p* = 0.043).

As inflammation may affect SPB, we conducted univariate logistic regression to analyze the relationship between inflammatory factors and SPB. The results are summarized in Appendix A. We finally selected the white blood cell count (WBC), platelet count (PLT), granulocyte ratio (GR), and platelet crit (PCT) as inflammatory confounders in the first trimester; and WBC, PLT, PCT, and neutrophil (NE) as inflammatory confounders in the third trimester. After adjustment for the inflammatory confounders, alone or combined with the clinical confounders, the association between the plasma Mn level in the third trimester and SPB remained (OR: 1.63, 95% CI: 1.04–2.55, *p* = 0.032; OR: 1.65, 95% CI: 1.04–2.62, *p* = 0.035).

### 3.4. Relationship between Mn Level in the Third Trimester and SPB Risk Stratified by BMI Grade

Stratified analyses were carried out based on BMI grade and parity. The results from the stratified analysis by BMI showed no significant difference between the SPB group and the controls (Table 4). Though a higher Mn concentration in the SPB group was found in the normal weight layer (1.30 vs. 1.15 ng/mL, *p* = 0.043; see Table 4), it was nonsignificant after Bonferroni correction (significance level α = 0.017 (0.05/3)). SPB risk was increased in the highest tertile Mn level in normal weight women (OR: 1.89, 95%CI: 1.12–3.20, *p* = 0.014). After adjustment for the clinical confounders, the SPB risk increased to 1.96 (95% CI: 1.13–3.38, *p* = 0.014). Moreover, a combined adjustment for clinical and inflammatory confounders made the risk association much more apparent (OR: 2.07, 95% CI: 1.18–3.61, *p* = 0.011) (see Table 5). However, there was no statistically significant relationship between the plasma Mn level and SPB risk when stratified by parity (Appendix A).

### 3.5. Relationship between Mn Level in the Third Trimester and SPB Risk in Non-PROM Women

Table 6 depicts the multiple logistic regression results. When it was limited to non-PROM participants, the SPB risk increased to 2.13 (95% CI: 1.11–4.10, *p* = 0.023) and 3.48 (95% CI: 1.84–6.57, *p <* 0.001) for the second and third tertile, respectively. Additionally, there was a clear dose-dependent relationship (*p* trend < 0.001) (see Figure 2). After a combined adjustment for clinical and inflammatory confounders, the SPB risk increased to 2.11 (95% CI: 1.05–4.22, *p* = 0.031) and 3.93 (95% CI: 2.00–7.74, *p* < 0.001) for the second and third tertile, respectively.

To test the robustness of our results, we conducted a sensitivity analysis by excluding women with vaginal GBS infection (*n* = 9 in the SPB group; *n* = 16 in the term labor group). The results did not change significantly (see Table 7).

## 4. Discussion

In this prospective nested case–control study, we identified for the first time that a high maternal plasma Mn level in the third trimester was positively associated with an increased risk of SPB, especially in normal-weight women and non-PROM women. Moreover, there is a dose-dependent relationship in the non-PROM population. The higher the level of plasma Mn in the third trimester, the higher the risk of SPB.

Solubilized Mn is absorbed in the intestine and transported across the microvilli to the blood via divalent metal transporter 1 (DMT1). They are absorbed in the form of Mn2+ or Mn3+, and Mn2+ is the most common form in the body [23,24]. In the blood, most Mn2+ is combined with albumin for transport and distribution [23]. Mn is distributed from plasma to the liver (accounting for 30% of total Mn), kidney (5%), pancreas (5%), colon (1%), urinary system (0.2%), bone (0.5%), brain (0.1%), red blood cells (0.02%), and the rest in soft tissue via DMT1 or transferrin receptor [23]. Most excessive Mn is combined with bile, transmitted into the intestine, and excreted. Trace Mn can also be detected in urine, sweat, and breast milk [23]. A study used rats as an animal model to explore Mn distribution during pregnancy. The results showed that the maternal liver accounts for most of the Mn, followed by the lung, brain, femur, and blood [25]. The distribution is similar to that of the non-pregnant subject. In addition, some of the Mn is distributed in the placenta and can be transported from maternal to fetal [25].

Blood Mn concentration, including whole blood and plasma/serum samples, is an effective and representative biomarker of Mn exposure [26,27]. Plasma/serum Mn accounts for 4% of the whole blood Mn concentration [27]. Most studies used whole blood samples for detection. The study results from different countries ranged from 10 to 22.5 ng/mL [4,7,16,28,29,30,31,32,33,34]. Several studies used plasma or serum for detection, ranging from 1.18 to 5.44 ng/mL [14,35,36,37,38]. In this study, the plasma Mn concentrations in both the first and third trimesters were significantly lower than those reported by the China Nutrition and Health Survey 2010–2012 of pregnant women (2.4 ng/mL, 95% CI: first trimester 1.1–9.2 ng/mL; third trimester 1.4–7.7 ng/mL) [39]. However, the plasma Mn concentrations in the third trimester observed in this study were similar to the Mn levels in the first trimester reported in a Shanxi and Anhui cohort study in China [14,15,37]. Due to the development of the coal mining industry, the environmental exposure risk of Mn is remarkably higher in Anhui and Shanxi provinces than in Beijing. Otherwise, our observed plasma Mn concentration in the third trimester was higher than that reported in an Indonesian cohort study (premature birth group: 1.09 ng/mL; term birth group: 1.0 ng/mL) [37]. Our results in the third trimester were similar to those reported in the Pakistan cohort study (1.38 ng/mL) [36]. The results may be affected by the participants’ age, BMI, gestational week, gravida, parity, sampling time, region, eating habits, laboratory testing methods, and environmental exposure. In addition, we found that there was a cumulative effect of the blood Mn levels during pregnancy, which was consistent with the research results in Taiwan, China [40], Wuhan [30], Anhui [37], and Costa Rica [4]. This may be related to the average growth and development of the fetus and the increased demand for the balance of the redox system during pregnancy. In addition, as it is necessary for bone formation [41], Mn concentration may increase in the third trimester for fetal growth. Since there is no clear evidence of a link between maternal Mn level and fetal development, further studies must be conducted.

Stratified by BMI grade, it showed a relationship between the two groups only in the normal weight layer. The reason may be that being overweight or thin is a risk factor for premature birth and would modify the effect of Mn on SPB. Researchers should fully evaluate these confounders when conducting further research. All the above findings suggested that the plasma Mn level in the third trimester may be associated with SPB, and the effect is significant in specific populations.

Several previous studies have reported that an excessive Mn concentration was associated with a higher preterm birth risk. Most studies did not strictly distinguish SPB from iatrogenic preterm birth, and most sampling times were at the time of delivery. Only one study conducted in a Shanxi cohort study estimated the relationship between the serum Mn level and SPB in the first and second trimesters. However, their serum samples in the first and second trimesters were from different populations. They reported a positive risk association between serum Mn levels and SPB in a positive dose-dependent manner in the first trimester, not the second trimester [14]. However, we found that a high concentration of plasma Mn (third tertile) in the third trimester of pregnancy, but not in the first trimester, was a risk factor for SPB. The serum Mn concentration in the first trimester of pregnant women in Shanxi was close to that in the third trimester in our study. Therefore, this may further indicate that a high concentration of Mn is associated with a higher risk of SPB, with no correlation to the pregnancy stage. In contrast, other previous studies did not find an association between the Mn concentration and preterm birth risk. Those inconsistent results may be related to population, study design, sample type, sampling time, and preterm birth type. Our study proved that the Mn concentration increases with gestational age when dynamically monitoring levels in pregnancy. However, most previous studies collected serum samples at delivery, whether there was a preterm birth (<37 weeks) or a term birth (≥37 weeks), resulting in different sampling times in the two groups. In our study, sampling times were similar in the SPB and term birth groups. Therefore, our results better clarify the relationship between a high maternal Mn concentration and SPB risk.

Considering that a vaginal GBS infection may impact SPB, we performed a sensitivity analysis after excluding women with the infection. The results showed that the relationship between the Mn concentration and SPB risk was still significant, further confirming the aforementioned results that a high Mn concentration is a risk factor for SPB.

The current understanding of the SPB mechanism is related to inflammation and oxidative stress [11], such as inflammatory factors, genetic variants, and predisposition [42]; alterations in inflammation and energy metabolism [43]; transcriptomics-determined chemokine-cytokine pathways [44]; telomeres and oxidative stress [45]; and so on. Mn toxicity is mediated, at least in part, by reactive oxygen species (ROS); depletion of cellular antioxidant defense mechanisms; and alterations in mitochondrial function and ATP production [12]. As part of metalloenzymes or their activators, Mn participates in maintaining the balance of the redox system in some critical physiological processes. An abnormal concentration of Mn may lead to premature birth by inducing an imbalance in the redox system in the body [46]. We hypothesized that higher levels of Mn during pregnancy might lead to oxidative stress caused by the imbalance between ROS and antioxidants. Higher levels of ROS may attack telomeres or another possible pathway, resulting in a higher risk of SPB eventually. All in all, the possible mechanism of the Mn effect on SPB needs further study.

There are some advantages to our study. First, we are the first to use a repeated measurement design to clarify the dynamic changes in plasma Mn during pregnancy in the related studies. Second, iatrogenic preterm birth would confuse the study results with a specific reason for preterm birth. In this study, only SPB cases were selected for analysis and strictly distinguished from iatrogenic preterm birth. Third, the sampling time was similar in the two groups, which could reduce the effect of the gestational week on element concentration. Fourth, this nested case–control study relied on a prospective cohort study conducted in Beijing; the same exposure backgrounds helped to reduce confounding in the region. Additionally, we were able to obtain the accurate information we required in the survey, which minimized recall bias.

Still, several limitations should be considered when explaining our results. First, we included no information on diet and nutrient supplementation, and we could not evaluate the pregnant women’s dietary Mn exposure level and its relationship with the plasma Mn concentration. Second, when the stratified analysis was carried out according to specific factors, some subgroups’ relatively small sample size may have affected the power to determine an association between the Mn level and SPB. Third, in this study, all cases were late-preterm (labor week ≥34 weeks), meaning they were close to full-term. The pathological condition related to SPB may be too small to observe a meaningful difference between the two groups. In future studies, it will be helpful to include early preterm women to clarify the relationship between plasma Mn levels and SPB. Fourth, parameters of oxidative stress were not assessed. As Mn may play an important role in SPB through oxidative stress, it would be better if the oxidative stress parameters were evaluated in our study. In future studies, we could further optimize the experimental design and try to assess related parameters of oxidative stress.

## 5. Conclusions

The increased plasma Mn concentration in the third trimester of pregnancy is associated with higher risks of SPB. The effect is much more significant in normal-weight women and non-PROM women. Our results provide a unique insight into SPB prevention in terms of monitoring and managing essential micronutrient elements during pregnancy. In addition, as the functioning of micronutrients especially tends to be in groups, further studies should be conducted to identify whether it is Mn alone or another compound that accompanies it that is associated with an increased risk of SPB.

## Figures and Tables

**Figure 1 nutrients-15-01413-f001:**
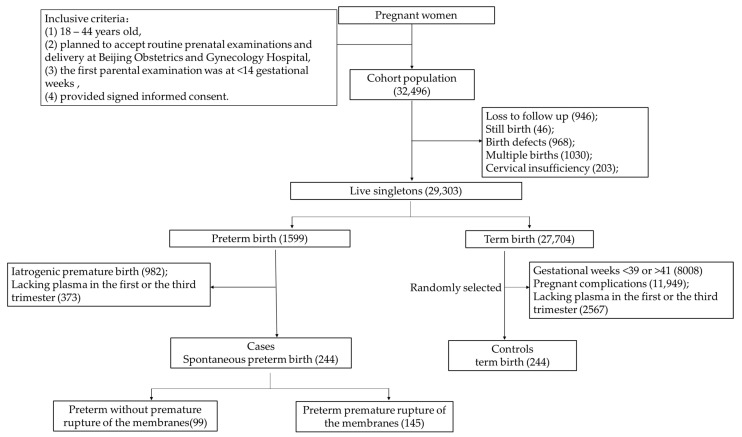
Flow diagram of the selection of cases and controls from the cohort.

**Figure 2 nutrients-15-01413-f002:**
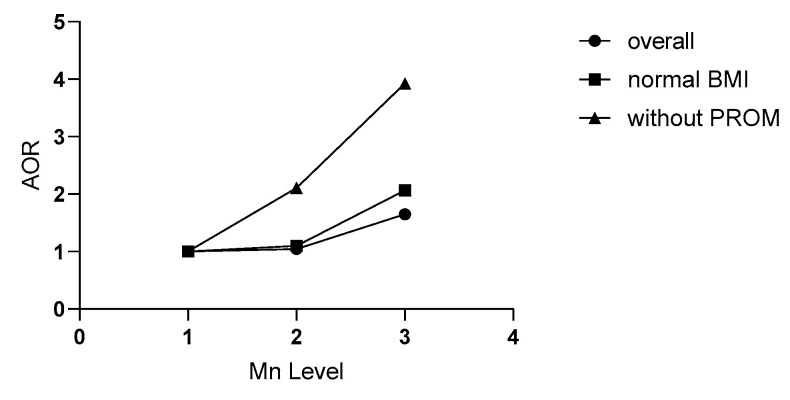
Relationships of maternal plasma Mn in the third trimester with SPB in different populations. Plasma Mn concentration was classified into three levels by the tertile of all subjects (L1, L2, and L3). Without PROM refers to the population without premature rupture of membranes (PROM); normal BMI refers to the people with a pre-pregnancy body mass index (BMI) in the 18.5–23.9 kg/m^2^ range. AOR refers to the odds ratio adjusted for age, BMI, education, economy, nationality, parity, gravida, fetal gender, blood week, and inflammatory factors related to SPB in routine blood tests. SPB: spontaneous preterm birth.

**Table 1 nutrients-15-01413-t001:** Characteristics of participants.

Characteristic	Total (N = 488)	SPB (N = 244)	TB (N = 244)	*p*
	*n* (%)	*n* (%)	*n* (%)	
Age (year) ^a^	31.51 ± 3.91	31.92 ± 3.95	31.09 ± 3.82	0.018
<30	161 (33)	76 (31.1)	85 (34.8)	0.128
30–34	212 (43.4)	101 (41.4)	111 (45.5)	
≥35	115 (23.6)	67 (27.5)	48 (19.7)	
BMI (kg/m^2^) ^a^	21.67 ± 2.90	22.14 ± 3.25	21.21 ± 2.4	<0.001
Slim	58 (11.9)	29 (11.9)	29 (11.9)	0.003
Normal	337 (69.1)	154 (63.1)	183 (75)	
Overweight	93 (19)	61 (25)	32 (13.1)	
Education				
Lower than bachelor’s	116 (23.8)	67 (27.5)	49 (20.1)	0.133
Bachelor’s	262 (53.7)	122 (50)	140 (57.4)	
Master’s or higher	110 (22.5)	55 (22.5)	55 (22.5)	
Economy				
<5000	19 (3.9)	11 (4.5)	8 (3.3)	0.513
5000–9999	92 (18.9)	50 (20.5)	42 (17.3)	
10,000–19,999	188 (38.6)	96 (39.3)	92 (37.9)	
≥20,000	188 (38.6)	87 (35.7)	101 (41.6)	
Nationality				
Han	449 (92)	223 (91.4)	226 (92.6)	0.617
Others	39 (8)	21 (8.6)	18 (7.4)	
Gravida				
1	239 (49)	111 (45.5)	128 (52.5)	0.124
≥2	249 (51)	133 (54.5)	116 (47.5)	
Parity				
Nulliparous	343 (70.3)	157 (64.3)	186 (76.2)	0.004
Multiparous	145 (29.7)	87 (35.7)	58 (23.8)	
GWG (kg) ^a^	13.82 ± 4.63	13.01 ± 4.70	14.63 ± 4.42	<0.001
Sampling time				
First trimester (week) ^b^	8 (1)	8 (1)	8 (1)	0.892
Third trimester (week) ^b^	34 (1)	34 (1)	34 (1)	0.824
Delivery way				
Natural birth	325 (66.6)	147 (60.2)	178 (73)	0.003
cesarean section	163 (33.4)	97 (39.8)	66 (27)	
Labor week (week) ^b^	37 (4)	36 (1)	40 (1)	<0.001
Fetal gender				
Boy	261 (53.5)	136 (55.7)	125 (51.2)	0.318
Girl	227 (46.5)	108 (44.3)	119 (48.8)	
Birth weight (g) ^a^	3067 ± 487	2701 ± 341	3434 ± 297	<0.001

SPB: spontaneous preterm birth; TB: term birth; BMI: body mass index; Slim: BMI < 18.5 kg/m^2^; Normal: 18.5 ≤ BMI < 23.9 kg/m^2^; overweight: BMI ≥ 24 kg/m^2^; GWG: gestational weight gain; *p* < 0.05 for statistical difference. ^a^ mean and standard deviation or ^b^ median and interquartile range for descriptive statistics.

**Table 2 nutrients-15-01413-t002:** Median Mn concentrations of women who had SPB and term birth.

	Median (IQR) (ng/mL)			*p* ^a^
	Total (*n* = 488)	SPB (*n* = 244)	TB (*n* = 244)	
First Trimester	0.81 (0.63)	0.82 (0.73)	0.79 (0.60)	0.441
Third Trimester	1.23 (0.67)	1.27 (0.66)	1.20 (0.61)	0.119
*p* ^b^	<0.001	<0.001	<0.001	

Mn: manganese; SPB: spontaneous preterm birth; TB: term birth; IQR: interquartile range; unit: *p <* 0.05 for statistical difference. ^a^ Mann–Whitney U test between cases and controls; *p <* 0.05 for statistical difference. ^b^ Wilcoxon signed rank test between the first and third trimester in total, cases, and control population.

**Table 3 nutrients-15-01413-t003:** Association of maternal plasma Mn levels with SPB.

Tertile Concentration of Mn (ng/mL)	Total	Cases	Controls	Crude OR	*p*	Adjusted OR ^a^	*p*	Adjusted OR ^b^	*p*	Adjusted OR ^c^	*p*
	*n* (%)	*n* (%)	*n* (%)	(95%CI)		(95%CI)		(95%CI)		(95%CI)	
First trimester											
<0.609	163 (33.4)	77 (31.6)	86 (35.2)	1		1		1		1	
0.609–1.012	163 (33.4)	82 (33.6)	81 (33.2)	1.13 (0.73–1.75)	0.58	1.12 (0.72–1.77)	0.612	1.09 (0.70–1.70)	0.716	1.123 (0.71–1.79)	0.614
>1.012	162 (33.2)	85 (34.8)	77 (31.6)	1.23 (0.80–1.91)	0.346	1.19 (0.76–1.87)	0.45	1.20 (0.76–1.87)	0.434	1.16 (0.73–1.85)	0.522
*p* trend				0.346		0.450		0.434		0.521	
Third Trimester											
>1.061	163 (33.5)	75 (30.7)	88 (36.2)	1		1		1		1	
1.061–1.470	162 (33.3)	77 (31.6)	85 (35)	1.06 (0.69–1.64)	0.784	1.08 (0.68–1.70)	0.754	1.06 (0.68–1.65)	0.804	1.04 (0.66–1.65)	0.862
>1.470	162 (33.3)	92 (37.7)	70 (28.8)	1.54 (1.00–2.39)	0.052	1.60 (1.02–2.53)	0.043	1.63 (1.04–2.55)	0.032	1.65 (1.04–2.62)	0.035
*p* trend				0.053		0.042		0.033		0.035	

Mn: manganese; SPB: spontaneous preterm birth; cases contain women with SPB; controls contain women with term birth; OR: odds ratio; CI: confidence interval. ^a^: unconditional logistic regression with adjustments for age, BMI, education, economy, nationality, parity, gravida, fetal gender, and sampling time. ^b^: unconditional logistic regression with adjustments for inflammatory factors related to SPB in the routine blood test. OR was adjusted by white blood cell count (WBC), platelet count (PLT), granulocyte ratio (GR), and platelet crit (PCT) in the first trimester, and adjusted by WBC, PLT, PCT, and neutrophil (NE) in the third trimester. ^c^: unconditional logistic regression with adjustment for age, BMI, education, economy, nationality, parity, gravida, fetal gender blood week, and inflammatory factors related to SPB in routine blood tests.

**Table 4 nutrients-15-01413-t004:** Mn concentrations of women who had SPB or term birth stratified by BMI grade.

	Median (IQR) (ng/mL)			
	Total	SPB	TB	
Slim				
*n* (%)	58 (11.9)	29 (11.9)	29 (11.9)	
First Trimester	0.77(0.63)	0.68(0.65)	0.95(0.66)	0.27
Third Trimester	1.29(0.78)	1.30(0.69)	1.20(0.91)	0.858
Normal				
*n* (%)	337 (69.1)	154 (63.1)	183 (75)	
First Trimester	0.80(0.63)	0.82(0.70)	0.78(0.62)	0.297
Third Trimester	1.22(0.68)	1.30(0.67)	1.15(0.59)	0.043
Overweight				
*n* (%)	93 (19.1)	61 (25)	32 (13.1)	
First Trimester	0.86(0.74)	0.88(0.80)	0.84(0.47)	0.701
Third Trimester	1.27(0.54)	1.26(0.67)	1.30(0.61)	0.282

BMI: body mass index; Slim: BMI < 18.5 kg/m^2^; Normal: 18.5 ≤ BMI < 23.9 kg/m^2^; Overweight: BMI ≥ 24 kg/m^2^. Mn: manganese; SPB: spontaneous preterm birth; TB: term birth; IQR: interquartile range; Bonferroni correction was used here and the adjusted significant *p*-value was 0.017 (0.05/3).

**Table 5 nutrients-15-01413-t005:** Association of maternal plasma Mn levels with SPB stratified by BMI grade.

Tertile Concentration of Mn (ng/mL)	Total	Cases	Controls	Crude OR	*p*	Adjusted OR ^a^	*p*	Adjusted OR ^b^	*p*	Adjusted OR ^c^	*p*
	*n* (%)	*n* (%)	*n* (%)	(95%CI)		(95%CI)		(95%CI)		(95%CI)	
Slim	58										
First trimester											
<0.609	23 (39.7)	13 (44.8)	10 (34.5)	1		1		1		1	
0.609–1.012	15 (25.9)	7 (24.1)	8 (27.6)	0.67 (0.18–2.49)	0.553	0.54 (0.09–3.27)	0.506	0.62 (0.16–2.48)	0.498	0.51 (0.07–3.60)	0.496
>1.012	20 (34.5)	9 (31)	11 (37.9)	0.63 (0.19–2.10)	0.452	0.39 (0.07–2.13)	0.279	0.81 (0.22–3.01)	0.758	0.66 (0.08–5.37)	0.699
*p* trend				0.446		0.280		0.736		0.699	
Third trimester											
<1.061	17 (29.3)	7 (24.1)	10 (34.5)	1		1		1		1	
1.061–1.470	19 (32.8)	10 (34.5)	9 (31)	1.59 (0.42–5.95)	0.493	3.87 (0.55–27.39)	0.175	1.98 (0.47–8.37)	0.352	5.28 (0.63–44.40)	0.125
>1.470	22 (37.9)	12 (41.4)	10 (34.5)	1.71 (0.48–6.16)	0.409	3.41 (0.65–17.92)	0.147	2.50 (0.61–10.34)	0.205	5.95 (0.84–42.20)	0.075
*p* trend				0.422		0.179		0.212		0.090	
Normal	337										
First trimester											
<0.609	116 (34.4)	48 (31.2)	68 (37.2)	1		1		1		1	
0.609–1.012	114 (33.8)	53 (34.4)	61 (33.3)	1.21 (0.72–2.04)	0.435	1.3 (0.76–2.23)	0.320	1.24 (0.73–2.11)	0.435	1.38 (0.79–2.41)	0.260
>1.012	107 (31.8)	53 (34.4)	54 (29.5)	1.39 (0.82–2.36)	0.222	1.33 (0.77–2.30)	0.304	1.32 (0.77–2.28)	0.318	1.27 (0.73–2.23)	0.400
*p* trend				0.221		0.297		0.314		0.387	
Third trimester											
<1.061	121 (36)	48 (31.2)	73 (40.1)								
1.061–1.470	104 (31)	44 (28.6)	60 (33)	1.12 (0.66–1.90)	0.688	1.17 (0.67–2.06)	0.572	1.09 (0.63–1.88)	0.760	1.10 (0.62–1.95)	0.747
>1.470	111 (33)	62 (40.3)	49 (26.9)	1.89 (1.12–3.20)	0.014	1.96 (1.13–3.38)	0.014	2.07 (1.20–3.54)	0.008	2.07 (1.18–3.61)	0.011
*p* trend				0.015		0.014		0.009		0.011	
Overweight	93										
First trimester											
<0.609	24 (25.8)	16 (26.2)	8 (25)	1		1		1		1	
0.609–1.012	34 (36.6)	22 (36.1)	12 (37.5)	0.92 (0.30–2.76)	0.877	0.99 (0.26–3.72)	0.984	0.79 (0.25–2.51)	0.692	0.69 (0.17–2.80)	0.604
>1.012	35 (37.6)	23 (37.7)	12 (37.5)	0.96 (0.32–2.88)	0.939	1.25 (0.33–4.76)	0.746	0.89 (0.28–2.78)	0.835	1.08 (0.26–4.55)	0.915
*p* trend				0.952		0.717		0.867		0.847	
Third trimester											
<1.061	25 (26.9)	20 (32.8)	5 (15.6)	1		1		1		1	
1.061–1.470	39 (41.9)	23 (37.7)	16 (50)	0.36 (0.11–1.16)	0.086	0.38 (0.10–1.41)	0.148	0.34 (0.10–1.16)	0.086	0.31 (0.08–1.26)	0.101
>1.470	29 (31.2)	18 (29.5)	11 (34.4)	0.41 (0.12–1.41)	0.156	0.32 (0.07–1.40)	0.129	0.36 (0.10–1.32)	0.123	0.18 (0.04–0.95)	0.044
*p* trend				0.188		0.144		0.153		0.046	

BMI: body mass index; Slim: BMI < 18.5 kg/m^2^; Normal: 18.5 ≤ BMI < 23.9 kg/m^2^; overweight: BMI ≥ 24 kg/m^2^. Mn: manganese; SPB: spontaneous preterm birth; cases contain women with SPB; controls contain women with term birth; OR: odds ratio; CI: confidence interval. ^a^: unconditional logistic regression with adjustment for age, BMI, education, economy, nationality, parity, gravida, fetal gender, and sampling time. ^b^: unconditional logistic regression with adjustment for inflammatory factors related to SPB in routine blood test. OR was adjusted by white blood cell count (WBC), platelet count (PLT), granulocyte Ratio (GR), and platelet crit (PCT) in the first trimester; and adjusted by WBC, PLT, PCT, and neutrophil (NE) in the third trimester. ^c^: unconditional logistic regression with adjustment for age, BMI, education, economy, nationality, parity, gravida, fetal gender blood week, and inflammatory factors related to SPB in routine blood tests. Bonferroni correction was used here, and the adjusted significant *p*-value was 0.017(0.05/3).

**Table 6 nutrients-15-01413-t006:** Association of maternal plasma Mn levels with SPB stratified by PROM.

Tertile Concentration of Mn (ng/mL)	Cases	Controls	Crude OR	*p*	Adjusted OR ^a^	*p*	Adjusted OR ^b^	*p*	Adjusted OR ^c^	*p*
	*n* (%)	*n* (%)	(95%CI)		(95%CI)		(95%CI)		(95%CI)	
First Trimester										
SPB with PROM										
<0.609	52 (35.9)	86 (35.2)								
0.609–1.012	50 (34.5)	81 (33.2)	1.03 (0.63–1.69)	0.895	1.04 (0.62–1.72)	0.949	1.00 (0.61–1.65)	0.990	1.04 (0.62–1.75)	0.941
>1.012	43 (29.7)	77 (31.6)	0.91 (0.55–1.51)	0.721	0.91 (0.54–1.54)	0.700	0.91 (0.54–1.51)	0.702	0.90 (0.53–1.52)	0.643
P trend			0.732		0.734		0.709		0.699	
SPB without PROM										
<0.609	25 (25.3)	86 (35.2)								
0.609–1.012	32 (32.3)	81 (33.2)	1.38 (0.75–2.52)	0.301	1.37 (0.73–2.57)	0.327	1.31 (0.70–2.44)	0.396	1.39 (0.72–2.66)	0.347
>1.012	42 (42.4)	77 (31.6)	1.85 (1.04–3.32)	0.038	1.72 (0.94–3.14)	0.084	1.80 (0.98–3.29)	0.058	1.71 (0.91–3.20)	0.098
*p* trend			0.036		0.077		0.056		0.092	
Third Trimester										
SPB with PROM										
<1.061	58 (40)	88 (36.2)								
1.061–1.470	42 (29)	85 (35)	0.75 (0.46–1.23)	0.255	0.75 (0.45–1.26)	0.319	0.76 (0.46–1.25)	0.28	0.74 (0.44–1.25)	0.303
>1.470	45 (31)	70 (28.8)	0.98 (0.59–1.61)	0.922	0.99 (0.59–1.67)	0.986	1.03 (0.62–1.71)	0.912	1.02 (0.60–1.73)	0.925
P trend			0.855		0.922		0.986		0.993	
SPB without PROM										
<1.061	17 (17.2)	88 (36.2)								
1.061–1.470	35 (35.4)	85 (35)	2.13 (1.11–4.09)	0.023	2.22 (1.12–4.39)	0.021	2.08 (1.07–4.05)	0.031	2.11 (1.05–4.22)	0.031
>1.470	47 (47.5)	70 (28.8)	3.48 (1.84–6.57)	<0.001	3.81 (1.96–7.40)	<0.001	3.69 (1.92–7.08)	<0.001	3.93 (2.00–7.74)	<0.001
*p* trend			<0.001		<0.001		<0.001		<0.001	

Mn: manganese; SPB: spontaneous preterm birth; PROM: premature rupture of membranes; cases contain SPB with PROM population or SPB without PROM population; controls contain women with term birth; all term birth women did not have PROM; OR: odds ratio; CI: confidence interval. Term delivery population as a reference. ^a^: unconditional multiple logistic regression with adjustments for age, BMI, education, economy, nationality, parity, gravida, fetal gender, and sampling time. ^b^: unconditional multiple logistic regression with adjustments for inflammatory factors related to SPB in routine blood test. OR was adjusted by white blood cell count (WBC), platelet count (PLT), granulocyte ratio (GR), and platelet crit (PCT) in the first trimester, and adjusted by WBC, PLT, PCT, and neutrophil (NE) in the third trimester. ^c^: Unconditional multiple logistic regression with adjustment for age, BMI, education, economy, nationality, parity, gravida, fetal gender blood week, and inflammatory factors related to SPB in routine blood tests. *p* < 0.05 for statistical significance.

**Table 7 nutrients-15-01413-t007:** Sensitivity analysis.

Tertile Concentration of Mn (ng/mL)	Total (*n* = 463)	Cases (*n* = 235)	Controls (*n* = 228)	Crude OR	*p*	Adjusted OR ^a^	*p*	Adjusted OR ^b^	*p*	Adjusted OR ^c^	*p*
	*n* (%)	*n* (%)	*n* (%)	(95%CI)		(95%CI)		(95%CI)		(95%CI)	
First Trimester											
<0.609	156 (33.7)	75 (31.9)	81 (35.5)								
0.609–1.012	153 (33)	77 (32.8)	76 (33.3)	1.09 (0.70–1.71)	0.692	1.10 (0.69–1.76)	0.680	1.04 (0.66–1.64)	0.879	1.10 (0.68–1.77)	0.697
>1.012	154 (33.3)	83 (35.3)	71 (31.1)	1.26 (0.81–1.97)	0.306	1.21 (0.76–1.92)	0.422	1.23 (0.78–1.95)	0.378	1.19 (0.74–1.92)	0.477
*p* trend				0.306		0.422		0.380		0.477	
Third Trimester											
<1.061	156 (33.8)	72 (30.6)	84 (37)								
1.061–1.470	151 (32.7)	74 (31.5)	77 (33.9)	1.12 (0.72–1.76)	0.617	1.13 (0.71–1.80)	0.615	1.10 (0.70–1.74)	0.671	1.08 (0.67–1.75)	0.738
>1.470	155 (33.5)	89 (19.3)	66 (29.1)	1.57 (1.01–2.46)	0.047	1.62 (1.01–2.57)	0.044	1.66 (1.05–2.63)	0.030	1.68 (1.04–2.69)	0.033
*p* trend				0.048		0.043		0.031		0.033	

Women with vaginal GBS infection were excluded from the sensitivity analysis. Mn: manganese; SPB: spontaneous preterm birth; GBS: group B streptococcus; cases contain women with SPB; controls contain women with term delivery; OR: odds ratio; CI: confidential interval. ^a^: unconditional logistic regression with adjustments for age, BMI, education, economy, nationality, parity, gravida, fetal gender, and sampling time. ^b^: unconditional logistic regression with adjustments for inflammatory factors related to SPB in routine blood test. OR was adjusted by white blood cell count (WBC), platelet count (PLT), granulocyte ratio (GR), and plateletcrit (PCT) in the first trimester, and adjusted by WBC, PLT, PCT, and neutrophil (NE) in the third trimester. ^c^: Unconditional logistic regression with adjustment for age, BMI, education, economy, nationality, parity, gravida, fetal gender blood week, and inflammatory factors related to SPB in routine blood tests. *p* < 0.05 for statistical significance.

## Data Availability

The data presented in this study are available on request from the corresponding author.

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
