# Peer review of "Association of Maternal Plasma Manganese with the Risk of Spontaneous Preterm Birth: A Nested Case–Control Study Based on the Beijing Birth Cohort Study (BBCS) in China"

_nutrients, 2023, doi:10.3390/nu15061413_

Round 1
Reviewer 1 Report
In their manuscript „Association of maternal plasma manganese with the risk of spontaneous preterm birth: A nested case-control study based on the Beijing Birth Cohort Study (BBCS) in China” (nutrients-2233836) the authors investigate in interesting Subject. Yet, several points need to be addressed, as outlined below.
Specific points
English editing by a native speaker is required (including removal of missing spaces and colloquial contractions). This should be clearly stated in the reply letter (please not just: “we did additional proofreading” or suchlike) and the person should be included in the acknowledgments.
Please consider that the study design allows only to detect correlations, but due to the nature of the data, no statements on causal relationships can be made. Therefore, such statements (e.g., “The highest Mn level (3 rd tertiles) in the third trimester increased the SPB risk…”) must be avoided throughout the entire manuscript.
Please provide a reference for the uptake of environmental Mn, especially that skin absorption is a noteworthy route of entry.
According to the introduction, 20 to 30% of preterm birth are iatrogenic. However, in the present cohort, it was over 60% (982 out of 1599). Was there any unusual influence in this cohort?
The authors indicate centrifugal force as “rpm”. As this is not a reproducible unit (varies with rotor diameter) it has to be replaced by “g”.
What kind of plasma has been used (anti-coagulant?). Were the samples tubes suitable for trace metal analysis?
Starting in line 115, the experimental procedure is written like an instruction, not a description.
Where was the In standard purchased? Concentration?
What were the purity and source of the nitric acid?
“in the SPB group were late preterm (34-36 +6 weeks), and the cesarean section rate was higher than that in the term labor group” was cesarean section not considered iatrogenic birth?
Table 2: Please add the unit on Mn concentration. Also for all subsequent tables where Mn concentrations are being indicated.
Why were different confounders for inflammation used in 1st and 3rd trimester? This seems arbitrary.
When the authors separate their data into subgroups based on multiple parameters, such as Mn level and BMI, patient numbers become rather small (down to 7 per group) making it questionable if this is still a representative number. Also, there is a lot of multiple testing by analyzing various subgroups and one wonders if correction for multiple testing might not affect the significance of the results (after all, most OR border on 1).
Section headers should not contain an interpretation of the results. For example, in 3.5 “There was a dose-dependent relationship between Mn level in the third trimester and SPB risk in non-PROM women.” should simply be “Relationship between Mn level in the third trimester and SPB risk in non-PROM women.”
Line 242: “The results may be affected by the participants' character”: How does one’s character affect Mn plasma levels?
In the paragraph starting in line 229, it seems as if the authors are comparing Mn levels in blood and plasma. As these are known to differ (erythrocytes contain the majority of Mn in blood) this is not a valid way to discuss Mn levels.
Line 274 is missing a closing bracket.
Author Response
Point 1: English editing by a native speaker is required (including removal of missing spaces and colloquial contractions). This should be clearly stated in the reply letter (please not just: "we did additional proofreading" or suchlike), and the person should be included in the acknowledgments.
Response 1: Thank you for the valuable comment. We are sorry for the poor English editing. We applied for a professional English-editing service for language and presentation editing. The corresponding changes have been marked up using the "Track Changes" function of MS Word. The proof has been submitted along with our revised manuscript. Also, the person who helped us edit our draft has been included in the acknowledgments.
Point 2: Please consider that the study design allows only to detect correlations, but due to the nature of the data, no statements on causal relationships can be made. Therefore, such statements (e.g., "The highest Mn level (3rd tertiles) in the third trimester increased the SPB risk…") must be avoided throughout the entire manuscript.
Response 2: Thank you for your valuable suggestion. We totally agree that our study design allows only to detect correlations rather than causal relationships. We have revised corresponding statements in the manuscript (line 28-29; line 213-214; line 237-239; line 331; line 344-345; line 401-402)
Point 3: Please provide a reference for the uptake of environmental Mn, especially that skin absorption is a noteworthy route of entry.
Response 3: Thank you for reminding us. We are sorry for the omission. We have added the reference for the uptake of environmental Mn in our revised manuscript. (line 56; Reference 10)
Point 4: According to the introduction, 20 to 30% of preterm birth are iatrogenic. However, in the present cohort, it was over 60% (982 out of 1599). Was there any unusual influence in this cohort?
Response 4: Thank you for reminding us. We have checked our dataset to confirm this issue. Beijing Obstetrics and Gynecology Hospital is a referral center for critical pregnant women in Beijing, China. A higher proportion of high-risk pregnant women are involved here, such as severe pre-eclampsia, placenta implantation, placenta previa, etc., resulting in a higher proportion of iatrogenic premature delivery.
Point 5: The authors indicate centrifugal force as “rpm”. As this is not a reproducible unit (varies with rotor diameter) it has to be replaced by “g”.
Response 5: Thank you for the valuable suggestion. We are sorry for the inappropriate description here. As it varies with rotor diameter, “rpm” is not reproducible. We have rechecked our original records and revised the centrifugal force into “1680g" in our revised manuscript. (line 134)
Point 6: What kind of plasma has been used (anti-coagulant?). Were the samples tubes suitable for trace metal analysis?
Response 6: Thank you for the comment. We are sorry for the confusing description here. We rechecked our original experiment record to confirm the type of plasma we used in the study. Maternal blood was collected in an anticoagulation tube lined with Ethylene Diamine Tetraacetic Acid (EDTA). Thus, the plasma contained anti-coagulant EDTA. We also replenished the above information in our revised manuscript. (line 133)
For some ethical reasons, all the blood samples we used in our study were residual samples initially collected for clinical testing. Due to the need for clinical testing, the sample tubes cannot be cleaned specifically. But the tubes used in the subsequent detection process were all cleaned with nitric acid. It may have little effect on testing results and statistical analysis.
Point 7: Starting in line 115, the experimental procedure is written like an instruction, not a description.
Response 7: Thank you for the comment. We are sorry for the inappropriate description here. We have revised the words according to the suggestion and embellished our language expression with the help of professionals. (line 137-141)
Point 8: Where was the In standard purchased? Concentration?
Response 8: Thank you for the comment. We are sorry for the omission here. We rechecked the purchase record and item instructions to confirm the information. In standard was purchased from the National Nonferrous Metals and Electronic Materials Analysis and Test Center in Beijing, China. The original concentration was 1000ug/ml. It would be diluted into 0.2ng/ml when used for determination. We have supplied the above information in our revised manuscript. (line138-139)
Point 9: What were the purity and source of the nitric acid?
Response 9: Thank you for the comment. We are sorry for the omission here. We verified the information according to our original experiment records. Nitric acid was purchased from Suzhou Crystal Clear Chemical Co., Ltd., And the purity was UPS grade (68%). The information was also added to our revised manuscript. (line139-140)
Point 10: “in the SPB group were late preterm (34-36 +6 weeks), and the cesarean section rate was higher than that in the term labor group” was cesarean section not considered iatrogenic birth?
Response 10: Thank you for the comment. We are sorry for the confusing expression here. Generally speaking, iatrogenic birth refers to the termination of pregnancy by medical intervention due to maternal or fetal factors unsuitable for continuous pregnancy. In our clinical practice, the main character distinguishing iatrogenic delivery from spontaneous delivery is whether the labor onset is spontaneous rather than delivery mode. Cesarean section meant to one of the delivery modes. Many factors may lead to cesarean delivery, such as macrosomia, fetal distress during the labor process, maternal request, and so on. Only part of cesarean is iatrogenic birth.
Point 11: Table 2: Please add the unit on Mn concentration. Also, for all subsequent tables where Mn concentrations are being indicated.
Response 11: Thank you for the valuable suggestion. We are sorry for the inapparent marking for the unit in the footnotes. We have added the unit of Mn concentration in the row in Tables according to the suggestion. (table2,3,4,5,6,7)
Point 12: Why were different confounders for inflammation used in 1st and 3rd trimester? This seems arbitrary.
Response 12: Thank you for the comment. We are sorry for the confusing expression here. As inflammation may affect SPB, we screened out the inflammatory factors in routine blood test that may affect SPB by univariate logistic regression analysis. The result was displayed in Table S1 in detail. The factors that may affect SPB were not exactly coincident in the 1st and 3rd trimester. Thus, confounders for inflammation were different in the two stages. We have revised the description in the Methods and results section. (line 173-175; line 219-224). In addition, we have fully referred to your suggestion and put all significant factors that screened out in different trimesters into adjusted mode simultaneously, and the results did not change.
Point 13: When the authors separate their data into subgroups based on multiple parameters, such as Mn level and BMI, patient numbers become rather small (down to 7 per group) making it questionable if this is still a representative number. Also, there is a lot of multiple testing by analyzing various subgroups and one wonders if correction for multiple testing might not affect the significance of the results (after all, most OR border on 1).
Response 13: Thank you for the comments to help us improve the analysis. We totally agree that small samples in some subgroups may affect the stability of the results and increase the risk of type II error (false negative). It is also one of our study's limitations, and we have pointed it out in the limitation section in our manuscript. The results need to be further verified by a larger population. (line 389-391)
Also, multiple testing in our study may affect the stability of the results and increase the risk of type I error (false positive). We really appreciated for you pointing out this to help us improve the analysis. We got through a literature review on the statistical approach, and Bonferroni correction were chosen for adjustment (Armstrong RA. Ophthalmic Physiol Opt. 2014). The significance level α was adjusted according to the number of multiple comparisons in order to reduce the false positive rate. We have revised description in the methods section (line 180-182), and updated our results in the revised manuscript. (line 234-237; line 326-328; Table 4; Table 5; Table S2; Table S3) In addition, as the possibility of false positive did exist, the results need to be further verified in future studies, especially in normal body mass index (BMI) pregnant women.
Point 14: Section headers should not contain an interpretation of the results. For example, in 3.5 "There was a dose-dependent relationship between Mn level in the third trimester and SPB risk in non-PROM women." should be "Relationship between Mn level in the third trimester and SPB risk in non-PROM women.”
Response 14: Thank you very much for the suggestion. We are sorry for the inappropriate expression here. We have revised all the section headers according to the suggestion. (line 198-199; line 208-209; line 231-232; line 249-251)
Point 15: Line 242: “The results may be affected by the participants' character”: How does one’s character affect Mn plasma levels?
Response 15: We apologize for the confusing description. "character" refers to one's age, BMI, gestational week, gravida, parity, etc. We have revised the description in our revised manuscript. (line313)
Point 16: In the paragraph starting in line 229, it seems as if the authors are comparing Mn levels in blood and plasma. As these are known to differ (erythrocytes contain the majority of Mn in blood) this is not a valid way to discuss Mn levels.
Response 16: Thank you for the comments. And we are sorry for the confusing description here. We have revised the description as follows: Blood Mn concentration, including the whole blood sample and plasma/serum sample, is an effective and representative biomarker of Mn exposure. Plasma/serum Mn accounts for 4% of the whole blood Mn concentration. Most studies used the whole blood samples for detection. The results showed a wide range from 10 to 22.5ng/ml. Several studies used plasma or serum for detection, and it ranges from 1.18 to 5.44 ng/ml. (line 295-300)
Point 17: Line 274 is missing a closing bracket.
Response 17: Thank you for your reminding us. We are sorry for the omission. We have corrected the mistake in the revised manuscript. (line 351)

Reviewer 2 Report
Comments to the Authors of manuscript number: nutrients-2233836 entitled “Association of maternal plasma manganese with the risk of spontaneous preterm birth: A nested case-control study based on the Beijing Birth Cohort Study (BBCS) in China”.
It is very interesting study involving very large group of participants. It is very time-consuming study, however it needs to be corrected before publication. Besides data presented here, it is worth to discuss the probably mechanism and Mn role in SPB.
1. L 43 – the abbreviation is introduced in the introduction for the first time, and should be explained
2. L 49, 50 - the reference should be added
3. L 53 – the sentence seems to be not finished
4. L 54- a systemic review is not adequate source, it should be original study or clinical
5. L 229 – the reference should be added and metabolism of this element should be described, how it is transported from intestine, how it is absorbed and distributed within the body, and whether there is a difference between the distribution in pregnant and nonpregnant subjects.
6. L 230- what Authors mean by “different regions”?
7. L 272, 283 – is it really dynamic?
8. L 294- The most manganese is found in the brain, liver, bones, kidneys and pancreas. Does its plasma concentration reflect its concentration in the body and availability?
It is necessary to bone formation, for this reason its amount increases in the 3th trimester. Why it is not discussed?
9. If it neutralizes free radical, why some parameters of oxidative stress are not assessed?
10. Why Authors did not give the daily demand of Mn for pregnant. Is it similar to nonpregnant, or differs between various age groups.
11. manganese reduces the absorption of other drugs and supplements, so if we use antibiotic treatment, we should take manganese 2 hours after taking the drug. Were there subjects treated with antibiotics?
12. Molecular mechanisms of Mn toxicity include oxidative stress, mitochondrial dysfunction, protein misfolding, endoplasmic reticulum (ER) stress, autophagy dysregulation, apoptosis, and disruption of other metal homeostasis. Why the possible mechanism of the preterm delivery is not given?
Author Response
It is very interesting study involving very large group of participants. It is very time-consuming study, however it needs to be corrected before publication. Besides data presented here, it is worth to discuss the probably mechanism and Mn role in SPB.
Response: Thank you very much for the valuable suggestion. We got through the literatures on Mn and SPB again and added related information in the introduction section and discussion section (line 60-64; 362-375)
Point 1: L 43 – the abbreviation is introduced in the introduction for the first time, and should be explained.
Response 1: Thank you for your reminding us. We are sorry for the omission. And the explanation and full name have been added to our revised manuscript. (line 45)
Point 2: L 49, 50 - the reference should be added
Response 2: Thank you for the suggestion. We are sorry for the omission. A related reference about the Mn exposure pathway has been added in the revised manuscript. (line 56; Reference 10).
Point 3: L 53 – the sentence seems to be not finished.
Response 3: Thank you for the comments. We are sorry for the unclear expression. We have revised the description. (line 66)
Point 4: L 54- a systemic review is not adequate source, it should be original study or clinical.
Response 4: Thank you for the comments. We are sorry for the inappropriate quotation here. In the subsequent sentences, we quoted original study results to support our opinion of the inconsistent effect on SPB. Thus, the inappropriate reference and description quoted from the systemic review could be deleted. We revised it in the revised manuscript. (line 66-68)
Point 5: L 229 – the reference should be added and metabolism of this element should be described, how it is transported from intestine, how it is absorbed and distributed within the body, and whether there is a difference between the distribution in pregnant and nonpregnant subjects.
Response 5: Thank you very much for the suggestion. It contributes to making the draft much more complete and higher quality. After getting through studies, we learned about part of the knowledge on Mn metabolism. Most of Mn was obtained from diet. Solubilized Mn is absorbed in the intestine and transported across the microvilli to the blood via divalent metal transporter 1 (DMT1). They are absorbed in the form of Mn2+ or Mn3+, and Mn2+ is the most common form in the body. In the blood, Most Mn2+ is combined with albumin in the blood for transport and distribution. Mn is distributed from plasma to the liver (accounting for 30% of total Mn), kidney (5%), pancreas (5%), colon (1%), urinary system (0.2%), bone (0.5%), brain (0.1%), red blood cell (0.02%), and the rest in soft tissue via DMT1 or transferrin receptor. Most excessive Mn is combined with bile and transmitted into the intestine and excrete out. Trace Mn can also be detected in urine, sweat and breast milk. (Roth JA., et al. Biol Res. 2006; Horning KJ, et al. Annu Rev Nutr.2015)
A study used rats as animal models to explore Mn distribution during pregnancy. The results showed that the maternal liver accounts for most of Mn; next are lung, brain, femur, and blood. The distribution is similar with the non-pregnant subject. In addition, some of the Mn is distributed in the placenta and can be transported from maternal to fetal through it. (Dorman DC, et al. Neurotoxicology. 2005)
We have also added the above information in our revised manuscript. (line282-294; Reference 22-24)
Point 6: L 230- what Authors mean by “different regions”?
Response 6: Thank you for the comment. We are sorry for the confusing description here. Different regions refer to different countries worldwide or different locations in one country. Countries involved are China, Japan, USA., Italy, Pakistan, Norway, and Indonesia. Also, it may refer to a different location in the same country, such as Anhui, Shanxi, and Wuhan province. They were located in the east, west, and south of China, respectively.
Point 7: L 272, 283 – is it really dynamic?
Response 7: Thank you for the comments. We are sorry for the confusing description, and we have some explanation for it. Each participant provided plasma samples twice during the whole gestation. The first time is at the first trimester of gestation (gestational weeks 8-10), and the second time is at the third trimester of gestation (gestational weeks 32-36). As for the two-period samples in the first and third trimester were from the same participants, the results would be well represented for dynamic change of Mn levels in pregnancy.
Point 8: L 294- The most manganese is found in the brain, liver, bones, kidneys and pancreas. Does its plasma concentration reflect its concentration in the body and availability?
It is necessary to bone formation, for this reason its amount increases in the 3th trimester. Why it is not discussed?
Response 8: Thank you for the comments. We totally agree that most manganese is distribute in tissues, such the liver, brain, bones, kidney, and so on. However, it would be impossible to study Mn level and its association with health in the human body using these tissues. The previous study proved that blood Mn level is an effective and representative biomarker of Mn exposure. (Marissa G., et al. J Occup Environ Hyg. 2014). Most previous studies chose the whole blood or plasma/serum samples for their accessibility. Thus, we selected plasma samples for the study. We added quotations here in the revised manuscript. (line 295-296; Reference 25-26)
Thank you very much for the comments. We greatly appreciate you pointing out this. It arouses a new view of data interpretation. Our results showed a higher Mn concentration in the third trimester than in the first trimester. As it is necessary for bone formation (alacios C. Crit Rev Food Sci Nutr. 200), Mn concentration may increase in the third trimester for fetal growth. It is coincident with our results. Since there is no clear evidence of maternal Mn level and fetal development, further studies can be conducted on it. We have also added the description in the revised manuscript. (line319-322; Reference 41)
Point 9: If it neutralizes free radical, why some parameters of oxidative stress are not assessed?
Response 9: Thank you for the comment. We totally agree that it would be better if the oxidative stress parameters were assessed in our study. It was one of the limitations of our study. For some ethical reasons, the plasma samples we used in our study were residual blood samples after clinical testing. The blood samples were relatively little. And there were no blood samples resting for oxidative stress parameters testing. In future studies, we could further optimize the experimental design and try to assess some of the parameters of oxidative stress. We also pointed out this in our limitations. (line 396-399)
Point 10: Why Authors did not give the daily demand of Mn for pregnant. Is it similar to nonpregnant, or differs between various age groups.
Response 10: Thank you for the comments. We are sorry for the information omission. We searched related literature and find the adequate intake (AI) criteria (Institute of Medicine (US) Panel on Micronutrients. Washington (DC): National Academies Press (US); 2001.) or recommended daily intake (RDI) criteria (WHO/FAO. Vitamin and Mineral Requirements in Human Nutrition, 2nd ed.; World Health Organization and Food and Agriculture Organization of the United Nations: Geneva, Switzerland, 2004). To reach an adequate intake of manganese during pregnancy, the AI suggested by the Institute of Medicine (US) is 2 mg/day. While in AHO/FAQ criteria, the RDI of Mn is 5mg/day. The recommendation was higher than that in non-pregnant women(1.8mg). There were no differs between various age groups from 14 to 50 years old during pregnancy. Also, the above information has been added in the introduction section in our revised manuscript. (line 50-53; Reference 8-9)
Point 11: manganese reduces the absorption of other drugs and supplements, so if we use antibiotic treatment, we should take manganese 2 hours after taking the drug. Were there subjects treated with antibiotics?
Response 11: Thank you for the comments. We are sorry for the unclear expression here. As most drugs would have negative effects on fetal development, it was rare for pregnant women to take drugs, including antibiotics, during pregnancy. We rechecked our dataset and confirmed that no participants were treated with antibiotics. We added the description in the results section in the revised manuscript. (line 195-196)
Point 12: Molecular mechanisms of Mn toxicity include oxidative stress, mitochondrial dysfunction, protein misfolding, endoplasmic reticulum (ER) stress, autophagy dysregulation, apoptosis, and disruption of other metal homeostasis. Why the possible mechanism of the preterm delivery is not given?
Response 12: Thank you very much for the comments. We totally agree that descriptions on the possible mechanism of preterm delivery would enhance our draft’s quality. We researched literature on the mechanism of SPB according to the suggestion. Labor (term and preterm) is characterized by increased myometrial contractility, cervical dilatation, and rupture of the chorioamnionitis membranes. The current understanding of the SPB mechanism is related to inflammation and oxidative, such as inflammatory factors, genetic variants, and predisposition; alterations in inflammation and energy metabolism, transcriptomics-determined chemokine-cytokine pathway; telomeres and oxidative stress; and so on. Mn toxicity is mediated, at least in part, by reactive oxygen species (ROS); depletion of cellular antioxidant defense mechanisms, and alterations in mitochondrial function and ATP production. We hypothesized that higher levels of Mn during pregnancy might lead to oxidative stress caused by the imbalance between ROS and antioxidants. Higher levels of ROS may attack telomeres or another possible pathway, resulting in a higher risk of SPB eventually. All in all, the possible mechanism of the Mn effect on SPB needs further studies. (Martinez-Finley EJ, et al. Free Radic Biol Med. 2013; Couceiro J, et al. Clin Genet. 2021; Wang P, et al. Am J Reprod Immunol. 2021; Phillippe M. et al. Am J Obstet Gynecol. 2022; Ischiropoulos H, et al. Int J Mol Sci. 2021)
The related information has been added in the introduction and discussion section in our revised manuscript. (line 60-64; line 362-375; Reference 11,12, 42-46)
Again we greatly appreciate the reviewer’s valuable comments, which are helpful in improving our manuscript.

Reviewer 3 Report
Thank you very much allowing me to review the article entitled “Association of maternal plasma manganese with the risk of A nested case-control study based on the Beijing Birth Cohort Study (BBCS) in China.” (nutrients-2233836) that is presented to NUTRIENTS for its possible publication in the Section “Micronutrients and Human Health”.
The aim of this study explores the relationship between maternal plasma Mn level and spontaneous preterm birth through repeated measurement data to find spontaneous preterm birth's underlying drivers in metal exposure and nutrition.
I attach some comments about this manuscript:
1.-When the incidence of 6.4% is indicated (today line 40) it must be indicated with the corresponding reference. The same thing happens in line 44 when talking about manganese and its role in fetal development. There are other statements that also need reference and introduction.
2.-In the introduction, the first time an acronym is mentioned, its meaning must appear: specifically in line 43 “SPB: spontaneous preterm birth”.
3.- In relation to the objective, it should be specified that the measurement is carried out in the first and third quarters.
4.- In the introduction, it would be very appropriate to have more information on the pathophysiological basis of how manganese can act on spontaneous preterm birth:.
5.- in material and methods it is indicated that it is a nested case-control study, on a prospective cohort; however, in line 78 it is indicated that the medical information was collected from the clinical history.
6.-in the calculation of the sample size, it assumes a risk of 2, what are they based on?
7.- The objective that appears at the end of the introduction talks about nutrition, but in material and methods there is no aspect related to nutrition that has been considered.
8.-differences in manganese levels in the first and third trimester appear both in the case group and in the control group, showing no significant differences between them. These results do not support the authors' conclusions, since the risks are not significant in the third trimester, although they are when adjusted. The authors should justify why they adjust for this variable, since, for example, fetal gender is not significant. in the initial table. Table 3 does not show statistical significance in the OR results.
9.-Table four identifies a significant difference in the third trimester between cases and controls. Have the authors considered that this is the most represented group in the sample and perhaps they identify the difference in the other group because the size is not appropriate?
10.- In table 5, 6, 7 the OR appears with 3 decimal places. I suggest that this is speculative and should be adjusted to two decimal places maximum, while the p-value should appear in all of them with 3 decimal places.
11.-I think that the authors should consider, especially in the discussion in the conclusions, the possible dryness of confusion because they only analyze manganese and other factors could be involved in this situation. Therefore, its conclusions should be more limited and raise the need for studies that assess whether it really is this micronutrient or is it one that accompanies it, since the functioning of micronutrients especially tends to be in groups.
Author Response
Point 1: When the incidence of 6.4% is indicated (today line 40) it must be indicated with the corresponding reference. The same thing happens in line 44 when talking about manganese and its role in fetal development. There are other statements that also need reference and introduction.
Response 1: Thank you for your comments. We are sorry for the information omission. We have added the related reference in our revised manuscript. (line 42; Reference 1-2; line 48; Reference 3-6)
Point 2: In the introduction, the first time an acronym is mentioned, its meaning must appear: specifically in line 43 “SPB: spontaneous preterm birth”.
Response 2: Thank you for reminding us. We are sorry for the omission and have revised it in the draft. (line 45)
Point 3: In relation to the objective, it should be specified that the measurement is carried out in the first and third quarters.
Response 3: Thank you very much for the suggestion. We have revised the description in the introduction section of our revised manuscript according to the suggestion. (line 81-82)
Point 4: In the introduction, it would be very appropriate to have more information on the pathophysiological basis of how manganese can act on spontaneous preterm birth.
Response 4: Thank you very much for the suggestion. We totally agreed that more information on the pathophysiological basis of how manganese can act on spontaneous preterm birth would make the draft higher quality. Labor (term and preterm) is characterized by increased myometrial contractility, cervical dilatation, and rupture of the chorioamnionitis membranes. The current understanding of the SPB mechanism is related to inflammation and oxidative, such as inflammatory factors, genetic variants, and predisposition; alterations in inflammation and energy metabolism, transcriptomics-determined chemokine-cytokine pathway; telomeres and oxidative stress; and so on. Mn toxicity is mediated, at least in part, by reactive oxygen species (ROS); depletion of cellular antioxidant defense mechanisms, and alterations in mitochondrial function and ATP production. We hypothesized that higher levels of Mn during pregnancy might lead to oxidative stress caused by the imbalance between ROS and antioxidants. Higher levels of ROS may attack telomeres or another possible pathway, resulting in a higher risk of SPB eventually. All in all, the possible mechanism of the Mn effect on SPB needs further studies. (Martinez-Finley EJ, et al. Free Radic Biol Med. 2013; Couceiro J, et al. Clin Genet. 2021; Wang P, et al. Am J Reprod Immunol. 2021; Phillippe M. et al. Am J Obstet Gynecol. 2022; Ischiropoulos H, et al. Int J Mol Sci. 2021). The related information has been added in the introduction section and discussion section in our revised manuscript. (line 60-64; 362-375)
Point 5: in material and methods it is indicated that it is a nested case-control study, on a prospective cohort; however, in line 78 it is indicated that the medical information was collected from the clinical history.
Response 5: Thank you for the comments. We are sorry for the confusing description. We collected the medical information prospectively when the participants came to the hospital for an antenatal examination. The information was recorded in the clinical system. When we need to analyze the data, we will leading-out the information from the clinical system. We have also revised the related description in the material and methods section in our revised manuscript (line 93-95)
Point 6: in the calculation of the sample size, it assumes a risk of 2, what are they based on?
Response 6: Thank you for the comments. We are sorry for the confusing expression here. We assumed the risk value as 2 based on a previous study conducted in Shanxi province, China. In that study, the author concluded that the SPB risk was significantly increased to 2.46 (95%CI: 1.08-5.62) in the highest level of Mn in the first trimester. We have revised the description and added a quotation in the calculation of the sample size section. (line153-154; Reference 14)
Point 7: The objective that appears at the end of the introduction talks about nutrition, but in material and methods there is no aspect related to nutrition that has been considered.
Response 7: Thank you for the comments. We are sorry for the confusing description. Our study aimed to find SPB's underlying drivers in micronutrient views. Mn, as a micronutrient, is obtained from the diet and absorbed from the intestine. Plasma Mn is an effective biomarker to reflect the nutrition intake of Mn at least partly. In our study, plasma samples were chosen for Mn level determination to reflect nutrition intake in a certain degree. We have also revised our description in the introduction and methods parts. (line 81; line 129-130)
In future study, we could optimize the study design and investigate the diet and supplementary intake related to Mn. We can assess the association of diet and supplemental Mn intake with plasma Mn level. Also, we can evaluate the relationship between diet and supplementary Mn intake and SPB risk. Lacking data on diet Mn intake was one of the limitations of our study, and we have pointed it out in the limitation section. (line 386-389)
Point 8: differences in manganese levels in the first and third trimester appear both in the case group and in the control group, showing no significant differences between them. These results do not support the authors' conclusions, since the risks are not significant in the third trimester, although they are when adjusted. The authors should justify why they adjust for this variable, since, for example, fetal gender is not significant. in the initial table. Table 3 does not show statistical significance in the OR results.
Response 8: Thank you for the comments. Though there was no significant difference in Mn levels between the two groups, we found a higher SPB risk in the highest tertile of Mn levels after being adjusted for confounders. According to the previous studies, age, BMI, education, economy, nationality, parity, gravida, fetal gender, and sampling time were all associated with SPB. Thus, we chose these factors as clinical confounders and put them into adjustment models. As inflammation may also affect SPB, we conducted a univariate logistic regression to find inflammatory confounders. In the final, white blood cell count (WBC), platelet count (PLT), Granulocyte Ratio (GR), and platelet crit (PCT) in the first trimester; and adjusted by WBC, PLT, PCT, and neutrophil (NE) in the third trimester were involved as inflammatory confounders. We have added some related quotations in the methods section of our revised manuscript. Besides, the P value has been supplied in the revised tables. (line 172-175; Reference 19-21; Table 3)
Point 9: Table four identifies a significant difference in the third trimester between cases and controls. Have the authors considered that this is the most represented group in the sample and perhaps they identify the difference in the other group because the size is not appropriate?
Response 9: Thank you for the comments. We totally agreed that small samples in some subgroups may affect the stability of the results and increase the risk of type II error (false negative). It is also one of our study's limitations, and we have pointed it out in the limitation section of our manuscript. The results need to be further verified by a larger population. (line 389-391)
Point 10: In table 5, 6, 7 the OR appears with 3 decimal places. I suggest that this is speculative and should be adjusted to two decimal places maximum, while the p-value should appear in all of them with 3 decimal places.
Response 10: Thank you very much for the suggestion. We have updated the tables in the revision draft. As all of the OR with 95%CI values should be adjusted to two decimal places maximum, we completely replaced tables with a new one. (Table 3, 5, 6, 7, S3)
Point 11: I think that the authors should consider, especially in the discussion in the conclusions, the possible dryness of confusion because they only analyze manganese and other factors could be involved in this situation. Therefore, its conclusions should be more limited and raise the need for studies that assess whether it really is this micronutrient or is it one that accompanies it since the functioning of micronutrients especially tends to be in groups.
Response 11: Thank you very much for the suggestions. We totally agree that other micronutrients, such as Fe, could also be involved in the situation because of the similar physiological function and metabolic mechanism. We have revised the description in the conclusion section and highlighted the need for further studies according to the suggestion. (line 405-408)
Again we greatly appreciate the reviewer’s valuable comments, which are helpful in improving our manuscript.

Round 2
Reviewer 1 Report
In the revised version of their manuscript „Association of maternal plasma manganese with the risk of spontaneous preterm birth: A nested case-control study based on the Beijing Birth Cohort Study (BBCS) in China” (nutrients-2233836) the authors have addressed all points raised in my previous evaluation. I have no further comments.
Reviewer 2 Report
I have no comments.
Reviewer 3 Report
Thank you very much allowing me to review the new version of the article entitled “Association of maternal plasma manganese with the risk of A nested case-control study based on the Beijing Birth Cohort Study (BBCS) in China.” (nutrients-2233836) that is presented to NUTRIENTS for its possible publication in the Section “Micronutrients and Human Health”.
I have carefully reviewed the clarifications made by the authors in the manuscript and the explanations in the reply letter.
I consider that the article has improved in understanding and it is a new contribution.
However, I believe that a direct causal relationship cannot be established, since although there is an association, the oxidative mechanism is very complex. However, I believe that this article raises a new line of research.